# Phosphatidylethanolamine-Binding Protein 1 Ameliorates Ischemia-Induced Inflammation and Neuronal Damage in the Rabbit Spinal Cord

**DOI:** 10.3390/cells8111370

**Published:** 2019-10-31

**Authors:** Woosuk Kim, Su Bin Cho, Hyo Young Jung, Dae Young Yoo, Jae Keun Oh, Goang-Min Choi, Tack-Geun Cho, Dae Won Kim, In Koo Hwang, Soo Young Choi, Seung Myung Moon

**Affiliations:** 1Department of Biomedical Sciences, and Research Institute for Bioscience and Biotechnology, Hallym University, Chuncheon 24252, Korea; tank3430@naver.com (W.K.); vovo1101@hallym.ac.kr (S.B.C.); 2Department of Anatomy and Cell Biology, College of Veterinary Medicine, and Research Institute for Veterinary Science, Seoul National University, Seoul 08826, Korea; hyoyoung@snu.ac.kr (H.Y.J.); vetmed2@snu.ac.kr (I.K.H.); 3Department of Anatomy, College of Medicine, Soonchunhyang University, Cheonan 31151, Korea; dyyoo@sch.ac.kr; 4Department of Neurosurgery, Hallym University Sacred Heart Hospital, College of Medicine, Hallym University, Pyeongchon 14068, Korea; ohjaekeun@gmail.com; 5Department of Thoracic and Cardiovascular Surgery, Chuncheon Sacred Heart Hospital, College of Medicine, Hallym University, Chuncheon 24253, Korea; gmchoi@hallym.or.kr; 6Department of Neurosurgery, Kangnam Sacred Heart Hospital, College of Medicine, Hallym University, Seoul 07441, Korea; jotak01@naver.com; 7Department of Biochemistry and Molecular Biology, Research Institute of Oral Sciences, College of Dentistry, Gangneung-Wonju National University, Gangneung 25457, Korea; kimdw@gwnu.ac.kr; 8Department of Neurosurgery, Dongtan Sacred Heart Hospital, College of Medicine, Hallym University, Hwaseong 18450, Korea; 9Research Institute for Complementary & Alternative Medicine, Hallym University, Chuncheon 24253, Korea

**Keywords:** phosphatidylethanolamine-binding protein 1, ischemia, oxidative stress, inflammation, spinal cord

## Abstract

In a previous study, we utilized a proteomic approach and found a significant reduction in phosphatidylethanolamine-binding protein 1 (PEBP1) protein level in the spinal cord at 3 h after ischemia. In the present study, we investigated the role of PEBP1 against oxidative stress in NSC34 cells in vitro, and ischemic damage in the rabbit spinal cord in vivo. We generated a PEP-1-PEBP1 fusion protein to facilitate the penetration of blood-brain barrier and intracellular delivery of PEBP1 protein. Treatment with PEP-1-PEBP1 significantly decreased cell death and the induction of oxidative stress in NSC34 cells. Furthermore, administering PEP-1-PEBP1 did not show any significant side effects immediately before and after ischemia/reperfusion. Administration of PEP-PEBP1 improved the Tarlov’s neurological score at 24 and 72 h after ischemia, and significantly improved neuronal survival at 72 h after ischemia based on neuronal nuclei (NeuN) immunohistochemistry, Flouro-Jade B staining, and western blot study for cleaved caspase 3. PEP-1-PEBP1 administration decreased oxidative stress based on malondialdehyde level, advanced oxidation protein products, and 8-iso-prostaglandin F2α in the spinal cord. In addition, inflammation based on myeloperoxidase level, tumor necrosis factor-α level, and high mobility group box 1 level was decreased by PEP-1-PEBP1 treatment at 72 h after ischemia. Thus, PEP-1-PEBP1 treatment, which decreases oxidative stress, inflammatory cytokines, and neuronal death, may be an effective therapeutic strategy for spinal cord ischemia.

## 1. Introduction

Interruption of the aorta or its segmental arteries results devastating and unpredictable complications, such as paraplegia caused by ischemia in spinal cord [1] during en bloc spondylectomy or aortic repair surgery [2,3]. Transient occlusion of aorta causes decreased blood flow in the spinal cord, and thus increases the cellular damage in the motor neurons of spinal cord [4]. Especially, rabbits receive homosegmental blood supply in the spinal cord, with no collateral blood supply coming from the thoracic spinal cord. In addition, the more caudal origin of the Adamkiewicz artery compared to that of other mammals is the rationale for using rabbits as an animal model for ischemia in the caudal half of the spinal cord [5]. Many have attempted to improve the prevention and treatment of spinal cord ischemia, but complications still affect the quality of life throughout the patient’s lifetime. Various cell death mechanisms have been identified in spinal cord ischemia, including oxidative stress by reactive oxygen species (ROS) after reperfusion and subsequent inflammation [6,7,8]. Neurons have many unsaturated fatty acids and utilize glucose as an energy source, and an interruption of blood supply and reoxygenation enormously increases the formation of ROS [9]. Phosphatidylethanolamine-binding protein 1 (PEBP1) has the ability to suppress the Raf1-mitogen activated protein kinase pathway, and therefore is also named Raf1 kinase inhibitory protein. PEBP1 stimulates acetylcholine synthesis in the median septal nuclei [10] and PEBP1 expression is decreased in hippocampus of Alzheimer patients [11]. In addition, chronic corticosterone treatment results in cognitive impairment via the down-regulation of PEBP1 expression in the hippocampus [12]. PEBP1 also promotes oligodendrocyte and neuronal differentiation [13,14]. Especially, PEBP1 overexpression facilitates the neuronal differentiation, although retinoic acid is not supplied into cell media [13]. In a previous proteomic approach, we observed that spinal cord ischemia significantly changes the various protein levels in the lumbar segments of spinal cord 3 h after ischemia in rabbits. Among differentially expressed proteins, we validated that PEBP1 was significantly decreased in the lumbar region 3 h after ischemia [15]. In addition, our colleagues observed that phosphorylated PEBP1 levels significantly increase in the hippocampal CA1 region 1–2 days after ischemia. In addition, administering PEP-1-PEBP1 significantly ameliorates the pyramidal cell damage induced by ischemia in gerbils via extracellular signal-regulated kinases and phosphorylation of cyclic-AMP response element binding protein signaling [16]. However, there are only few studies on the effects of PEBP1 against ischemic damage in the spinal cord.

In the previous study, we generated a PEP-1-PEBP1 fusion protein and confirmed it could penetrate the blood-brain barrier and plasma membrane of neurons [16]. In the present study, we examined the effects of PEP-1-PEBP1 treatment on H_2_O_2_-induced oxidative stress in NSC34 cells in vitro and ischemic damage in the rabbit spinal cord in vivo.

## 2. Materials and Methods

### 2.1. PEP-1-PEBP1 Fusion Protein Construction and Assessment Its Efficiency

#### 2.1.1. Cell Preparation

NSC34 cells were used in the present study and cells were maintained in the conditions described in the previous study [15]. Special medium was utilized to promote the neuronal differentiation as mentioned by Eggett et al. [17], and the cell had motor neuron-like morphology at seven days after growing [17].

#### 2.1.2. Construction of PEP-1-PEBP1

The cDNA sequence for human PEBP1 was amplified by polymerase chain reaction (PCR) using the PEBP1-specific sense primer 5′-CTC GAG ATG CCG GTG GAC C-3′, which contained an *Xho*I restriction site, and the PEBP1-specific anti-sense primer, 5′-GGA TCC CTA CTT CCC AGA CAG C-3′, which contained a *Bam*HI restriction site. PEP-1-PEBP1 was by ligation of a TA cloning vector with human PEBP1 cDNA and expression plasmid was inserted into a pET-15 vector with His-PEP-1-PEBP1 or His-PEBP1. PEP-1-PEBP1 and control-PEBP1 proteins were produced as described in a previous study [16], and the production of PEP-1-PEBP1 and control-PEBP1 proteins was confirmed by western blot analysis for polyhistidine antibody (1:2000, His-probe, Santa Cruz Biotechnology) as described previously [16,18]. 

#### 2.1.3. Transduction of PEP-1-PEBP1 Proteins into NSC34 Cells

Transduction of PEP-1-PEBP1 proteins into NSC34 cells was confirmed as described previously [15]. Briefly, NSC34 cells received dose-dependent (0.5–3 μM) treatment of PEP-1-PEBP1 or control-PEBP1 protein and harvested 1 h after treatment. In addition, cells were treated with 3 μM PEP-1-PEBP1 and control-PEBP1 treatment and harvested at various times (0–36 h) after treatment to observe the intracellular penetration efficacy and stability of PEP-1-PEBP1 or control-PEBP1 protein. Expression of PEP-1-PEBP1 and control-PEBP1 proteins was evaluated by western blot analysis for polyhistidine antibody, as described previously [16,18].

#### 2.1.4. Intracellular Location of PEP-1-PEBP1

Intracellular delivery of PEP-1-PEBP1 or control-PEBP1 protein was confirmed by immunocytochemical staining for polyhistidine, as described previously [15,16]. Briefly, cells cultured on coverslips were incubated with 3 μM PEP-1-PEBP1 or control-PEBP1 protein for 1 h at 37 °C, and the cells were fixed with 4% paraformaldehyde for 5 min. The immunocytochemical procedures were conducted as described previously [16], and the cells were subsequently incubated with rabbit anti-polyhistidine antibody (1:2000, SantaCruz Biotechnology, Santa Cruz, CA, USA) and AlexaFluor 488-conjugated anti-rabbit IgG (1:1500, Invitrogen, Carlsbad, CA, USA) at 25 °C for 1 h with 4′,6-diamidino-2-phenylindole (Roche Applied Science, Mannheim, Germany).

### 2.2. Protective Effects of PEP-1-PEBP1 against Oxidative Stress in NSC34 Cells

#### 2.2.1. DNA Damage and Oxidative Stress

To elucidate the effects of PEP-1-PEBP1 or control-PEBP1 protein on DNA damage and oxidative stress, terminal deoxynucleotidyl transferase dUTP nick-end labeling (TUNEL) and 2′,7′-dichlorofluorescein diacetate (DCF-DA) staining was done as described in previous studies [15,16]. Briefly, NSC34 cells were incubated with 3 μM PEP-1-PEBP1 or control-PEBP1 protein for 1 h and thereafter exposed to 1 mM hydrogen peroxide (H_2_O_2_) for 3 h and 10 min for TUNEL and DCF-DA staining as described previously [15], respectively. Thereafter, the cells were treated with 20 μM DCF-DA for 30 min for DCF-DA staining. TUNEL and DCF-DA staining were observed with a fluorescence microscope (Nikon eclipse 80i, Tokyo, Japan) and staining intensity of DCF-DA was evaluated by a Fluoroskan enzyme-linked immunosorbent assay (ELISA) plate reader (Labsystems Oy) at 485 nm excitation and 538 nm emission wavelengths. In addition, fluorescence positive cells were counted under a phase-contrast microscopy (×200 magnification) [19,20].

#### 2.2.2. Measurement of Cell Viability

Effects of PEP-1-PEBP1 or control-PEBP1 protein on cell viability was assessed based on a WST-1 assay kit (Daeillab Service, Seoul, South Korea) as described previously [15]. Briefly, PEP-1-PEBP1 or control-PEBP1 protein (0.5–3 μM) was pretreated in NSC34 cells for 1 h and thereafter cells were exposed to 1 mM H_2_O_2_ for 5 h. Cell viability was measured at 450 nm using an ELISA microplate reader (Labsystems Multiskan MCC/340, Helsinki, Finland), and cell viability was expressed as a percentage value versus that in the untreated control cells.

#### 2.2.3. Western Blot Analysis

Equal amounts of cell lysates were resolved by 12% sodium dodecyl sulfate-polyacrylamide gel electrophoresis (SDS-PAGE) and the proteins were transferred to a polyvinylidene difluoride (PVDF) membrane. The membranes were blocked with 5% nonfat dry milk in TBST buffer (25 mM Tris-HCl, 140 mM NaCl, 0.1% Tween 20, pH 7.5). Then, the membrane was probed with the rabbit anti-caspase 3 (CST #9662S, Cell Signaling Technology, Beverly, MA, USA), and rabbit anti-cleaved caspase 3 (c-caspase 3, CST #9661S, Cell Signaling Technology), and horse anti-rabbit IgG (Vector, Burlingame, CA, USA). The proteins were detected using enhanced chemiluminescent (ECL; Amersham, Buckinghamshire, UK) protocol according to the manufacturer’s instructions [19].

### 2.3. In vivo Neuroprotective Action of PEP-1-PEBP1 against Spinal Cord Ischemia

#### 2.3.1. Experimental Animals

Male New Zealand white rabbits (1.2–1.5 kg) were purchased from Experimental Animal Center (Cheonan Yonam College, Cheonan, South Korea) and were housed as described in the previous studies [15,21]. The Institutional Animal Care and Use Committee (IACUC) of Seoul National University approved the animal procedures (SNU-160613-18). 

#### 2.3.2. Determination of Effective PEP-1-PEBP1 Concentration against Ischemic Damage

To observe dose-dependent effects of PEP-1-PEBP1 on motor neurons in ventral horn of spinal cord after ischemia, the animals were divided into five groups: Sham-operated (control), PEP-1 peptide, 10 mg/kg control-PEBP1, 1 mg/kg PEP-1-PEBP1, 3 mg/kg PEP-1-PEBP1, and 10 mg/kg PEP-1-PEBP1 treated groups. Animals received treatment of PEP-1 peptide, control-PEBP1, or PEP-1-PEBP1 immediately after reperfusion.

#### 2.3.3. Ischemic Surgery 

The ischemic surgery was performed as described in a previous study [15]. Briefly, animals were anesthetized with 2.5% isoflurane (Baxter, Deerfield, IL, USA) and occluded the abdominal aorta with an aneurysm clip at subrenal region for 30 min. Body temperature of rabbits were tightly controlled (38.7 ± 0.3 °C) using a thermometric blanket during anesthesia. Sham operation was also conducted for control group with same procedures without occlusion of abdominal aorta. 

#### 2.3.4. Monitoring of Physiological Parameters

Blood in all groups was obtained from ear vein to measure the blood glucose, pH, and arterial blood gases (PaO_2_ and PaCO_2_) before ischemic surgery and 10 min after reperfusion, as described previously [15]. Mean arterial pressure (MAP) was assessed in caudal artery by physiography (Physiograph; Gould Instrument Systems, OH, USA) and acquisition software (Ponemah version 3.0; Gould Instrument Systems).

#### 2.3.5. Neurological Assessment

Neurological function of animals was assessed by modified Tarlov criteria, as described in previous studies [15,21], with two independent observers for each experiment 24 h and 72 h after reperfusion because the animals showed neurological deficits 12–24 h after reperfusion [22,23].

#### 2.3.6. Detection of Neuronal Survival and Cell Death

Rabbits (*n* = 5 in each group) were anesthetized with 2 g/kg urethane (Sigma) after the neurological assessment and perfused transcardially, as described previously [15,21]. Lumbar segments (L_5_-L_6_) of spinal cord were removed and 30-μm-thick sections were obtained using a cryostat (Leica, Wetzlar, Germany).

Immunohistochemical staining for neuronal nuclei (NeuN) was conducted as described previously [15,21]. Sections were subsequently incubated with a mouse anti-NeuN antibody (1:1000; Millipore, Temecula, CA, USA), biotinylated goat anti-mouse IgG, followed by a streptavidin-peroxidase complex (1:200, Vector). Immunoreactive structures were visualized by reaction with 3,3′-diaminobenzidine tetrahydrochloride in 0.1 M Tris-HCl buffer (pH 7.2).

The number of NeuN-immunoreactive cells in all the groups were counted using an image analysis system (software: Optimas 6.5^®^, CyberMetrics, Scottsdale, AZ, USA) as described previously [15,21]. 

To investigate the degeneration/death of cells, Fluoro-Jade B (FJB, a fluorescent marker for the localization of cellular degeneration) histofluorescence staining was conducted according to the method published by Candelario-Jalil et al. [24]. In brief, the sections were immersed in 1% sodium hydroxide in 80% alcohol and followed in 70% alcohol. They were then transferred to 0.06% potassium permanganate solution and incubated in 0.0004% FJ B (Histochem, Jefferson, AR, USA) solution. Finally, they were placed on a slide warmer (about 50 °C) to be reacted. The reacted sections were examined using an epifluorescent microscope (Carl Zeiss, Göttingen, Germany), which was equipped with blue excitation light (450–490 nm).

#### 2.3.7. Biochemical Assessments in Spinal Cord Tissue

To measure biochemical parameters in spinal cord tissue, control, PEP-1 peptide-treated, 10 mg/kg Control-PEBP1-treated, and 3 mg/kg PEP-1-PEBP1-treated rabbits (*n* = 5 in each group) were euthanized with overdose of urethane (Sigma) 72 h after reperfusion, and spinal cord tissue at L_5_-L_6_ levels were obtained. Quantitative analysis was conducted by western blot analysis for caspase 3 and c-caspase 3 in the spinal cord. Briefly, animals were sacrificed using 2 g/kg of the anesthetic urethane (Sigma-Aldrich). Lumbar segments (L_5_-L_6_) of spinal cord were removed and used for western blot study as described in a previous study [25]. Briefly, the protein-transferred membrane was sequentially incubated with rabbit anti-caspase 3 (1:1000, Cell Signaling Technology) or rabbit anti-c-caspase 3 (1:1000, Cell Signaling Technology), peroxidase-conjugated goat anti-rabbit IgG (1:1000, Vector), and an ECL chemiluminescent kit (Pierce; Thermo Fisher Scientific, Inc., Waltham, MA, USA).

Tissue MDA (Cayman Chemical Company, Ann Arbor, MI, USA), MPO (Cusabio, Hubei, China), HMGB (IBL, Hamburg, Germany), TNF-α (R&D Systems Inc., Minneapolis, MN, USA), and 8-iso-PGF2α (Cayman Chemical Company) levels were measured by commercially available ELISA kits. AOPP levels were measured by a spectrophotometric method (Schimadzu UV 1601 spectrophotometer) in the presence of potassium iodide at 340 nm as demonstrated by Witko-Sarsat et al. [26] and calibrated with chloramine-T solutions. The AOPP levels were expressed in micromoles chloramine-T equivalents per liter.

### 2.4. Statistical Analysis

Data were shown as mean with standard errors of mean or 95% confidence interval and analyzed statistically using by Student *t*-test or a one-way analysis of variance (ANOVA), followed by Bonferroni’s post-hoc test using GraphPad Prism 5.01 software (GraphPad Software, Inc., La Jolla, CA, USA).

## 3. Results

### 3.1. Production and Delivery of PEP-1-PEBP1 Protein into NSC34 Cells

PEP-1-PEBP1 was manufactured by fusion of human *PEBP1* gene and a PEP-1 expression vector (Figure 1A). Following overexpression in yeast, purification of PEP-1-PEBP1 and control-PEBP1 proteins were conducted with a Ni^b+^- → Ni^2+^- nitrilotriacetic acid Sepharose affinity column and PD-10 column chromatography. Western blot analysis with a polyhistidine antibody identified PEP-1-PEBP1 and control-PEBP1 proteins at approximately 23 kDa and 25 kDa, confirming the successful expression of these proteins (Figure 1B).

Concentration- and time-dependent changes in polyhistidine expression were determined in NSC-34 cells. Weak polyhistidine expression was found at 1 μM PEP-1-PEBP1 treatment and expression increased dose-dependently by 3 μM PEP-1-PEBP1 treatment. However, polyhistidine expression was not observed upon treatment with any concentration of control-PEBP1 (Figure 2A). In addition, polyhistidine expression was found at 30 min upon treatment with 3 μM PEP-1-PEBP1, and expression increased with time by 1 h after treatment. Similarly, no polyhistidine expression was found at any time after control-PEBP1 treatment (Figure 2B).

The intracellular stability of transduced PEBP1 protein in NSC34 cells was assessed by western blotting for polyhistidine. Polyhistidine protein levels had decreased 24 h after PEP-1-PEBP1 treatment, and had disappeared at 36 h after treatment (Figure 2C). 

Transduced PEBP1 was observed by polyhistidine immunocytochemistry, and no signals were observed in the control NSC34 cells or cells 1 h after 3 μM control-PEBP1 treatment. In contrast, treatment with 3 μM PEP-1-PEBP1 showed polyhistidine immunoreactivity in the cytoplasm and cell membrane, but not in the nucleus of NSC34 cells (Figure 2D).

### 3.2. Effects of PEP-1-PEBP1 on Oxidative Damage in NSC34 Cells

Exposure to 1 mM H_2_O_2_ for 3 h significantly decreased cell viability to 47.96% of the viability of the control group. However, pre-treatment with PEP-1-PEBP1 protein for 1 h before exposure to H_2_O_2_ showed dose-dependent increases in cell viability, and 3 μM PEP-1-PEBP1 showed significant increases in cell viability compared to that in the control-PEBP1-treated group, and cell viability was 66.37% of the viability of the control group (Figure 3A). However, treatment with various concentrations of control-PEBP1 did not significantly change cell viability.

ROS formation was measured by DCF-DA-fluorescent staining in NSC34 cells. Treatment with 1 mM H_2_O_2_ for 3 h showed strongly DCF-DA positive cells, and reactive fluorescence intensity was significantly increased compared to that in the control group. Pretreatment with control-PEBP1 did not significantly change DCF-DA reactive fluorescence in NSC34 cells compared to that in the H_2_O_2_-treated vehicle group. However, PEP-1-PEBP1 pretreatment significantly decreased DCF-DA reactive fluorescence compared to that in the H_2_O_2_-treated vehicle group (Figure 3B).

Apoptotic cell death was assessed by TUNEL staining 5 h after exposure to H_2_O_2_ in NSC34 cells. Exposure to 1 mM H_2_O_2_ for 5 h showed abundant TUNEL-positive cells, and reactive fluorescence intensity was significantly increased compared to that in the control group. Pretreatment with PEP-1-PEBP1 before exposure to 1 mM H_2_O_2_ significantly decreased the number of TUNEL-positive cells compared to that in the H_2_O_2_-treated vehicle group, while pretreatment with control-PEBP1 showed a similar number of TUNEL-positive cells with that in the H_2_O_2_-treated vehicle group (Figure 3C).

Cell death was also confirmed 5 h after exposure to H_2_O_2_ in NSC34 cells by western blot analysis for caspase 3 and c-caspase 3. Caspase 3 protein levels did not show any significant changes in all groups after control-PEBP1 or PEP-1-PEBP1 treatment, while c-caspase 3 protein levels were dramatically increased in vehicle group after H_2_O_2_ exposure compared to that in the control group. Treatment with control-PEBP1 had no effects on the c-caspase 3 levels in NSC34 cells compared to that in the vehicle group, but PEP-1-PEBP1 treatment significantly reduced the c-caspase 3 protein levels by 46.98% of vehicle group in NSC34 cells (Figure 3D).

### 3.3. Effects of PEP-1-PEBP1 on Physiological Data against Spinal Cord Ischemia 

The acute side effects of control-PEBP1 or PEP-1-PEBP1 treatment on physiological conditions such as blood pH, pO_2_, pCO_2_, and glucose levels, as well as distal MAP, were measured before ischemic surgery and 10 min after reperfusion. Administration of control-PEBP1 or PEP-1-PEBP1 did not significantly change these physiological parameters before ischemic surgery and 10 min after reperfusion (Table 1). Effects of PEP-1-PEBP1 were evaluated with Tarlov’s neurological score 24 h and 72 h after reperfusion, NeuN immunohistochemical staining 72 h after reperfusion. In the control group, all rabbits did not show any neurological problems in posture and behavior during experimental periods. In addition, abundant NeuN-positive nuclei were detected in the ventral horn of spinal cord. In the vehicle-treated and control-PEBP1-treated groups, rabbits showed abnormal hindlimb posture with a mean neurological score of 0.75–0.80 at 24 h and 72 h after reperfusion. In addition, few NeuN-positive nuclei were detected in the ventral horn, and the number of NeuN-positive nuclei was 11.89% and 14.27% of that in the control group, respectively. In the 1 mg/kg PEP-1-PEBP1-treated group, the Tarlov’s neurological scores were slightly increased compared to that in the vehicle (PEP-1 peptide)-treated group, and some NeuN-positive nuclei were detected in the ventral horn, although the difference in the number of NeuN-positive nuclei was not statistically significant between control and vehicle-treated groups. In the 3 and 10 mg/kg PEP-1-PEBP1-treated groups, rabbits were able to move voluntarily and showed higher neurological scores of 2.75–3.25 at 24 h and 72 h after reperfusion. Many NeuN-positive nuclei were observed in the ventral horn in the 3 and 10 mg/kg PEP-1-PEBP1-treated groups, and the number of NeuN-positive nuclei was 65.30% and 74.00% of that of the control group (Figure 4).

### 3.4. Effects of PEP-1-PEBP1 on Neuronal Death against Spinal Cord Ischemia 

The effects of control-PEBP1 or PEP-1-PEBP1 treatment on neuronal cell death were evaluated by FJB staining and western blot for caspase 3 and c-caspase 3 in the spinal cord. In control group, FJB positive cells were not detected in the ventral horn of spinal cord. In the PEP-1 peptide- and control-PEBP1-treated groups, FJB positive cells were found in the ventral horn of spinal cord and c-caspase 3 was prominently increased compared to that in the control group. In these groups, there were significant differences on FJB positive cells and c-caspase levels in the spinal cord. In the PEP-1-PEBP1-treated group, fewer FJB positive cells were found in the spinal cord compared to that in the PEP-1 peptide- or control-PEBP-1-treated group, respectively. In addition, c-caspase 3 levels were also significantly decreased in the PEP-1-PEBP1-treated group compared to that in the PEP-1 peptide-treated group (Figure 5). 

### 3.5. Effects of PEP-1-PEBP1 on Oxidative Stress and Inflammation after Spinal Cord Ischemia

In spinal cord tissues, levels of MDA, AOPP, and 8-iso-PGF2α, three oxidative stress biomarkers, were measured to determine ischemia-related oxidative stress in the spinal cord 72 h after ischemia. In the PEP-1 peptide-treated group, MDA, AOPP, and 8-iso-PGF2α levels were significantly increased in the spinal cord after ischemia by 225.04%, 145.64%, and 1049.17% of that of the control group, respectively. In the control-PEBP1-treated group, MDA and AOPP levels were similarly detected compared to those in the PEP-1 peptide-treated group, although 8-iso-PGF2α level was slightly decreased in the control-PEBP1-treated group. However, the administration of PEP-1-PEBP1 significantly reduced ischemia-induced elevations of MDA, AOPP, and 8-iso-PGF2α levels to 126.43%, 103.77%, and 402.18% of that of the control group, respectively (Figure 6). 

In the PEP-1 peptide-treated and control-PEBP1-trateed groups, MPO, TNF-α, and HMGB levels were significantly robustly increased in the spinal cord 72 h after ischemia compared to those in the control group. In this group, MPO, TNF-α, and HMGB levels in the spinal cord homogenates were 284.97%, 949.70%, and 471.92% of those in the control group, respectively. In the control-PEBP1-treated group, MPO level was similar to that in the PEP-1 peptide-treated groups, while TNF-α and HMGB levels were slightly decreased in the control-PEBP1-treated group compared to that in the PEP-1 peptide-treated group, respectively. Administration of PEP-1-PEBP1 ameliorated these dramatic increases of MPO, TNF-α, and HMGB levels in the spinal cord to 188.10%, 456.68%, and 203.31% of that of the control group, respectively (Figure 6). 

## 4. Discussion

PEBP1 is involved in neuronal cell death and cognitive deficits in neurodegenerative diseases [27]. In addition, PEBP1 expression was shown to be decreased in various neurological disorders such as stroke [28], mild cognitive impairments [29], and Alzheimer’s disease [30]. However, other studies demonstrated that PEBP1 expression is increased in the spinal cord 24 h after contusion damage [31] and in the penumbra tissue 6 h after focal ischemia [32]. However, our previous proteomic study demonstrated that PEBP1 expression was decreased in the spinal cord 3 h after ischemia [15], and we also observed PEP-1-PEBP1 protected pyramidal neurons from transient forebrain ischemia in gerbils [16]. In the present study, we investigated the neuroprotective potentials of PEP-1-PEBP1 against spinal cord ischemia using NSC34 cells in vitro and a spinal cord ischemia model using rabbits in vivo.

In the present study, we observed concentration- and time-dependent delivery of PEP-1-PEBP1 proteins in the NSC34 cells, which have similar morphological and physiological properties as motor neurons [33,34,35]. However, treatment with control-PEBP1 did not result in polyhistidine expression in any concentration or time after treatment. This result was consistent with results of our previous study that PEP-1-PEBP1 was efficiently transduced into HT22 cells derived from the hippocampus [16]. In addition, we previously demonstrated that PEP-1-hemo oxygenase-1 fusion protein efficiently delivered into the ventral horn of rabbit spinal cord [36]. In the present study, we also observed the stable expression of PEP-1-PEBP1 by 24 h after treatment and transduced PEBP1 expression in the cytoplasm of NSC34 cells 1 h after PEP-1-PEBP1 treatment. This result was supported by previous studies showing that PEBP1 is mainly expressed in the cytoplasm and plasma membrane [16,37,38,39].

In this study, we observed the effects of control-PEBP1 and PEP-1-PEBP1 on H_2_O_2_-induced oxidative damage in NSC34 cells because glutamate-induced excitotoxicity has no specific effects in NSC34 cells [40]. Treatment with PEP-1-PEBP1, but not with control-PEBP1, significantly reduced cell death and ROS formation in NSC34 cells after exposure to H_2_O_2_ based on caspase 3 western blot, TUNEL, DCF-DA staining, and WST-1 assays. This result suggests that PEP-1-PEBP1 prevents neuronal death from oxidative damage via reducing ROS formation. Depletion of PEBP1 significantly increases generation of mitochondrial superoxide in HEK-499 cells [41]. In contrast, overexpressing PEBP1 or administering PEP-1-PEBP1 ameliorates the generation of mitochondrial superoxide in HEK-499 cells [35] and ROS formation in HT22 cells [16].

We examined the neuroprotective effects of control-PEBP1 or PEP-1-PEBP1 against ischemic damage in the rabbit spinal cord. Acute side effects of control-PEBP1 or PEP-1-PEBP1 10 min before and after ischemia were monitored, but we did not observe any significant change in physiological parameters such as pH, pO_2_, pCO_2_, or glucose levels. Spinal cord ischemia showed neurological deficits in rabbits’ hindlimb functions and posture judging from Tarlov’s criteria at 24 h and 72 h after ischemia/reperfusion. Administering control-PEBP1 resulted in similar neurological deficits in rabbits compared to that in the PEP-1 peptide-treated group at 24 h and 72 h after ischemia/reperfusion. In addition, NeuN-immunoreactive surviving neurons were few in the PEP-1-peptide or control-PEBP1 treated groups compared to that in the control group at 72 h after ischemia/reperfusion, while FJB positive cells and c-caspase 3 protein levels were significantly increased compared to that in the control group. However, the administration of PEP-1-PEBP1 showed dose-dependent improvements in rabbits’ neurological scores at 24 h and 72 h after ischemia/reperfusion, although the neurological scores were slightly decreased at 72 h after ischemia. In addition, administration of PEP-1-PEBP1 significantly ameliorated the increases of FJB positive cells and c-caspase expression in the spinal cord induced transient ischemia. Spinal cord ischemia is caused by aortal occlusion during aortic repair surgery [1], and it would be easy to prevent ischemia-induced neuronal damage by administering PEP-1-PEBP1 to patients undergoing aortic repair surgery. In addition, protein therapy with PEP-1-PEBP1 can be applicable immediately if desired, while overexpression of PEBP1 by lentiviral vector requires enough time to express the PEBP1. Electroacupuncture at the acupoints of the governor vessel in animals with spinal cord injury significantly changes the expression of PEBP1 in the spinal cord [42]. Depletion of PEBP1 suppresses the neuronal differentiation of SH-SY5Y neuroblastoma cells induced by retinoid acid, while PEBP1 overexpression facilitates differentiation into neurons without retinoic acid treatment in media [13,14]. In addition, PEBP1 also promotes the differentiation of hippocampal precursor cells following hypoxic ischemia [43]. This result is consistent with previous studies reporting that overexpressing PEBP1 or administering PEP-1-PEBP1 ameliorates ischemia-induced neuronal damage in brain ischemia models [16,28]. 

Spinal cord membranes consist of polyunsaturated fatty acids, and ischemia-induced ROS causes lipid peroxidation with MDA as a decomposition product [44]. AOPP is closely related to protein oxidation such as cross-liked protein products that contain dityrosine [45]. In addition, 8-iso-PGF_2α_ has been considered a specific and sensitive indicator for peroxidation caused by ROS [46]. In the present study, we measured MDA, AOPP, and 8-iso-PGF_2α_ levels to observe ischemia-induced lipid peroxidation and protein oxidation in the spinal cord. Spinal cord ischemia significantly elevated the MDA, AOPP, and 8-iso-PGF_2α_ levels in the spinal cord at 72 h after ischemia/reperfusion, consistent with results of a previous study [47]. In the present study, administration of control-PEBP1 did not show any significant decreases in MDA, AOPP, and 8-iso-PGF_2α_ levels in the spinal cord at 72 h after ischemia compared to that in the PEP-1 peptide-treated group. In contrast, PEP-1-PEBP1 significantly ameliorated increases of MDA, AOPP, and 8-iso-PGF_2α_ levels in the spinal cord following ischemia, suggesting that administering PEP-1-PEBP1 reduces ischemia-induced lipid peroxidation and protein oxidation in the spinal cord.

MPO is a hemoprotein expressed by polymorphonuclear neutrophils and is an indicator of neutrophil infiltration into tissues [48]. In the present study, we observed increased MPO levels in the spinal cord at 72 h after ischemia compared to that in the control group. This result is supported by a previous study that found MPO activity is significantly increased in the spinal cord ischemic group compared to that in the sham group [49]. We observed a significant reduction of MPO levels in the spinal cord by PEP-1-PEBP1, not control-PEBP1, treatment following ischemia/reperfusion. In addition, we also observed TNF-α levels in the spinal cord because ischemia recruits neutrophils, which mediate the inflammatory response by producing TNF-α and other cytokines [50,51]. Spinal cord ischemia significantly increased the TNF-α levels in the spinal cord compared to that in the control group and administering PEP-1-PEBP1, not control-PEBP1, mitigated the ischemia-induced elevation of TNF-α levels in the spinal cord after ischemia/reperfusion. This result is supported by previous studies reporting that spinal cord ischemia robustly increases TNF-α levels [52,53]. HMGB1 is another inflammatory mediator that injured cells immediately release the HMGB1 to establish and facilitate inflammatory responses [54], and inhibiting HMGB1 release can be a novel therapeutic approach to treat spinal cord injury [55]. In the present study, we observed increases of HMGB1 levels in the spinal cord after ischemia/reperfusion. This result is consistent with results of a previous study that HMGB1 mRNA and protein levels were significantly increased in the spinal cord after ischemia/reperfusion [56]. In the present study, administering PEP-1-PEBP1 ameliorated ischemia-induced release of HMGB1 in the spinal cord compared to that in the PEP-1 peptide- or control-PEBP1-treated groups, suggesting that inhibiting HMGB1 release may be one of the mechanisms for PEBP1 to reduce neuronal damage induced by ischemia. Electroacupuncture also reduces ischemic neuronal damage by inhibiting HMGB1 release in the spinal cord [56]. 

In conclusion, we generated a PEP-1-PEBP1 fusion protein and observed the effective penetration of PEP-1-PEBP1 into NSC34 cells. In addition, we observed the neuroprotective effects of PEP-1-PEBP1 against H_2_O_2_-induced oxidative stress in NSC34 cells as well as ischemia-induced neuronal death in rabbit spinal cord, reducing inflammation and oxidative damage. These results suggest that the PEP-1-PEBP1 fusion protein may be applicable to reduce neuronal damage during aortic repair surgery and en bloc spondylectomy by suppressing oxidative stress and inflammation.

## Figures and Tables

**Figure 1 cells-08-01370-f001:**
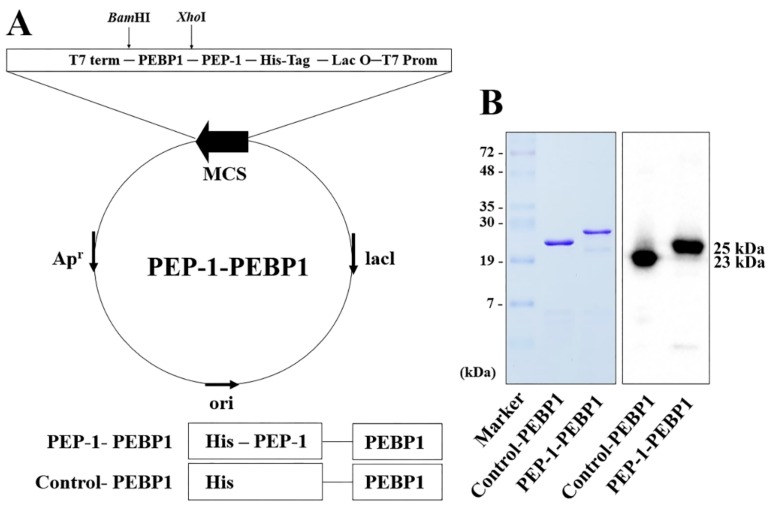
Purification and expression of control-PEBP1 and PEP-1-PEBP1 fusion protein in NSC34 cells. (**A**) Generation of control-PEBP1 and PEP-1-PEBP1 protein. (**B**) Western blot analysis for polyhistidine showing the successful purification and expression of control-PEBP1 and PEP-1-PEBP1 proteins.

**Figure 2 cells-08-01370-f002:**
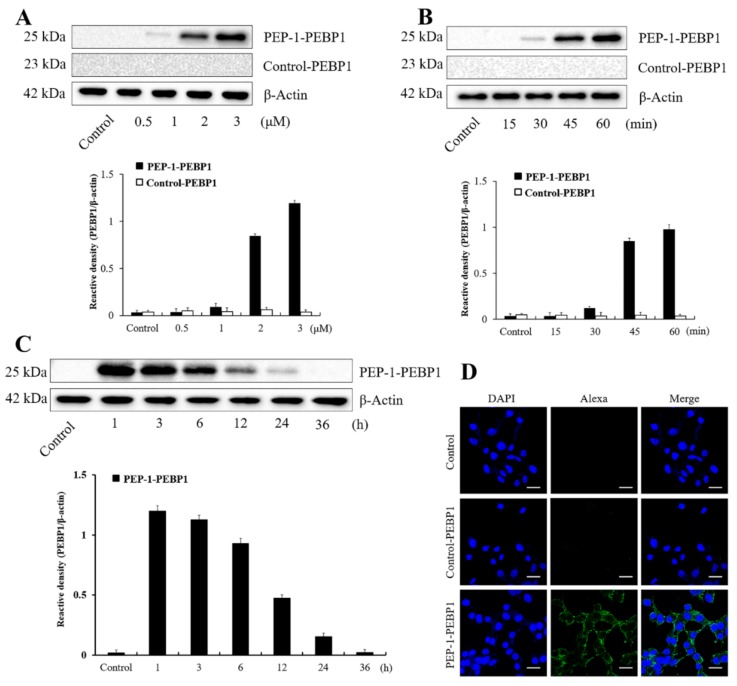
Transduction efficacy and stability of control-PEBP1 and PEP-1-PEBP1 fusion protein in NSC34 cells. (**A**) Western blot analysis showing concentration-dependent (0.5–3 μM) cellular expression of polyhistidine for 1 h after PEP-1-PEBP1 and control-PEBP1 treatment. (**B**) Western blot analysis showing time-dependent (0–60 min) cellular expression of polyhistidine, analyzed after 3 μM PEP-1-PEBP1 and control-PEBP1 treatment. (**C**) Western blot analysis showing chronological (1–36 h) intracellular stability of polyhistidine expression for 1 h after PEP-1-PEBP1 treatment. (**D**) Immunocytochemical staining for polyhistidine in PEP-1-PEBP1 and control-PEBP1 protein for 1 h after PEP-1-PEBP1 treatment. Scale bar = 20 μm. The bars indicate mean ± SEM.

**Figure 3 cells-08-01370-f003:**
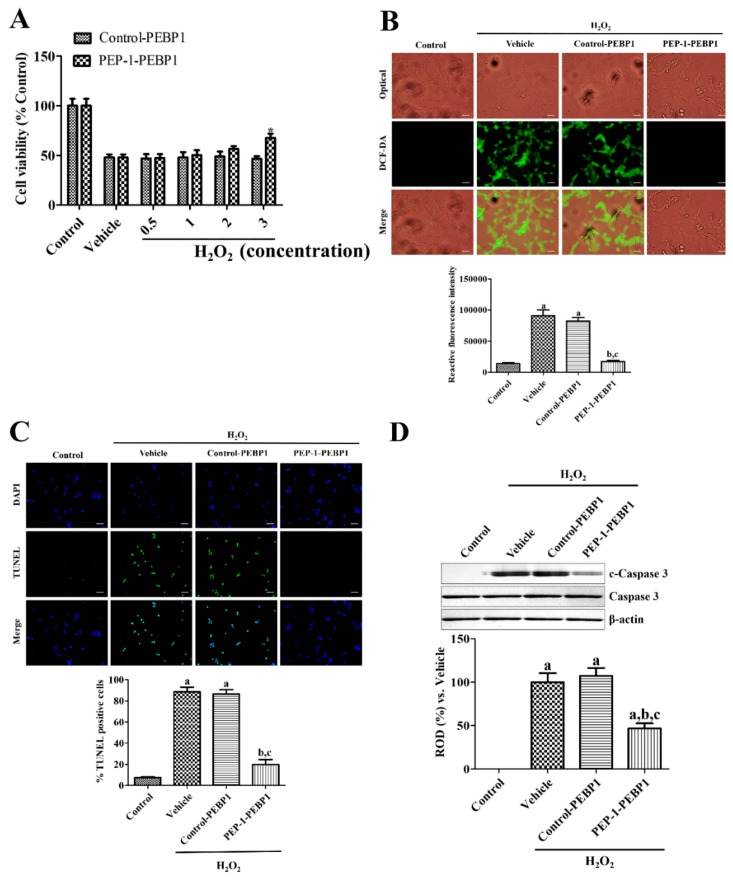
Protective effects of PEP-1-PEBP1 and control-PEBP1 proteins on oxidative damage induced by 1 mM H_2_O_2_ treatment in NSC34 cells. (**A**) ELISA assay results for dose-dependent (0.5–3 μM) viability of NSC34 cells analyzed by WST-1 assay. (**B**) Immunofluorescent staining for DCF-DA and measurement of ROS production using a fluorescence plate reader. (**C**) Immunofluorescent staining for TUNEL and quantitative analysis of TNINEL positive cells by cell counting under a phase-contrast microscopy. Scale bar = 50 μm (B and C). (**D**) Western blot analysis for caspase 3 and cleaved caspase 3 (c-caspase 3) in the NSC34 lysates. Data were analyzed by Student *t*-test (* *p* < 0.05, significantly different from the control-PEBP1-treated group) or one-way analysis of variance followed by a Bonferroni’s post-hoc test (^a^
*p* < 0.05, significantly different from the control group; ^b^
*p* < 0.05, significantly different from the vehicle-treated group; ^c^
*p* < 0.05, significantly different from the control-PEBP1-treated group). The bars indicate mean ± SEM.

**Figure 4 cells-08-01370-f004:**
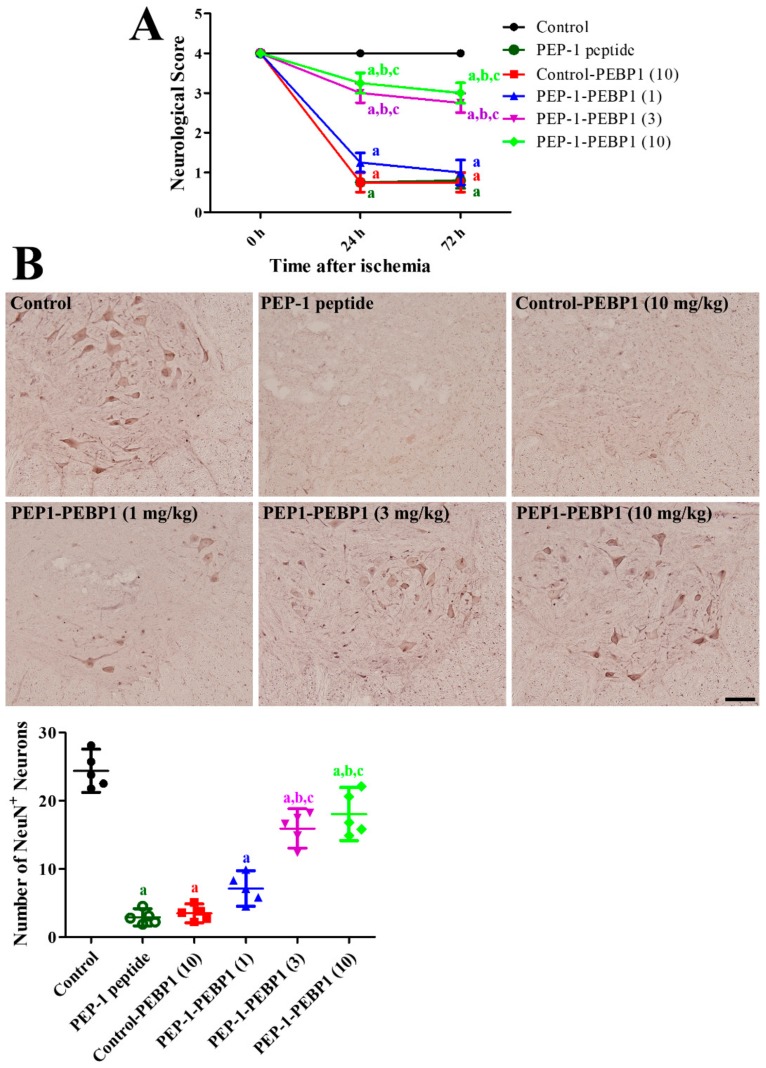
Effect of PEP-1-PEBP1 protein against ischemic damage in the rabbit spinal cord. (**A**) Neurological deficits are assessed by Tarlov’s criteria 24 h and 72 h after reperfusion (*n* = 10 per group; ^a^
*p* < 0.05, significantly different from the control group, ^b^
*p* < 0.05, significantly different from the PEP-1 peptide-treated group; ^c^
*p* < 0.05, significantly different from the control-PEBP1-treated group). The bars indicate SEM. (**B**) Immunohistochemistry for NeuN to label the surviving neurons in the ventral horn of the spinal cord 72 h after ischemia/reperfusion. The number of NeuN-positive nuclei per section in all the groups is also shown (*n* = 5 per group; ^a^
*p* < 0.05, significantly different from the control group, ^b^
*p* < 0.05, significantly different from the PEP-1 peptide-treated group; ^c^
*p* < 0.05, significantly different from the control-PEBP1-treated group). The bars indicate confidence interval or standard errors of mean.

**Figure 5 cells-08-01370-f005:**
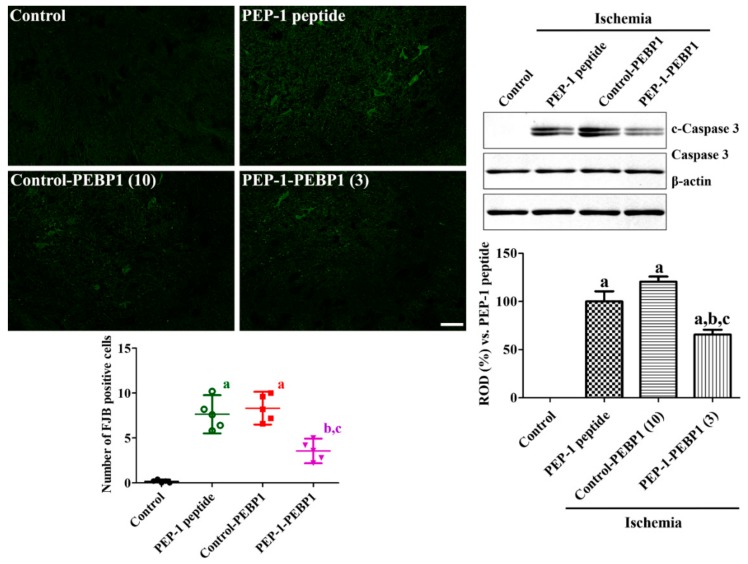
Effect of Control-PEBP1 and PEP-1-PEBP1 protein on cell death and caspase 3 expression in the rabbit spinal cord against ischemic damage. Fluoro-Jade B (FJB) staining is conducted to detect the degenerating cells in the spinal cord 72 h after reperfusion. The number of FJB-positive cells in all the groups is also shown. Western blot analysis for caspase 3 and cleaved caspase 3 (c-caspase 3) in the spinal cord homogenates is conducted to compare the caspase 3 dependent neuronal death in spinal cord (*n* = 5 per group; ^a^
*p* < 0.05, significantly different from the control group, ^b^
*p* < 0.05, significantly different from the PEP-1 peptide-treated group; ^c^
*p* < 0.05, significantly different from the control-PEBP1-treated group). The bars indicate confidence interval or standard errors of mean.

**Figure 6 cells-08-01370-f006:**
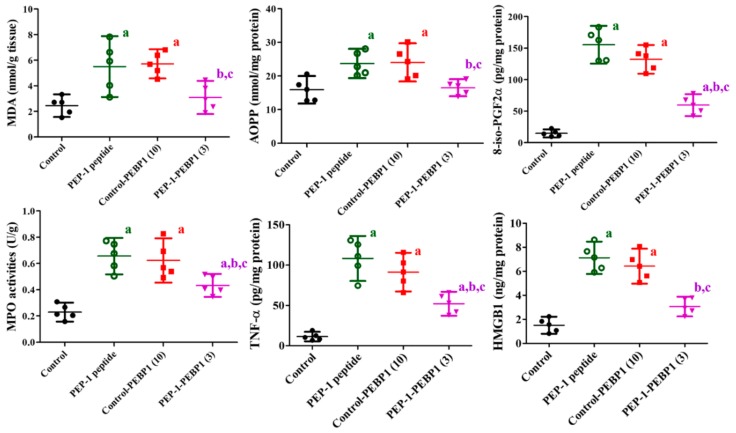
Effects of Control-PEBP or PEP-1-PEBP1 on MDA, AOPP, 8-iso-PGF2α, MPO, TNF-α, and HMGB in the spinal cord 72 h after ischemia/reperfusion (*n* = 5 per group; ^a^
*p* < 0.05, significantly different from the control group, ^b^
*p* < 0.05, significantly different from the PEP-1 peptide-treated group, ^c^
*p* < 0.05, significantly different from the control-PEBP1-treated group). The bars indicate confidence interval.

**Table 1 cells-08-01370-t001:** Physiological parameters 10 min before and after ischemic surgery.

	pH	Distal MAP (mmHg)	PaCO_2_ (mmHg)	PaO_2_ (mmHg)	Glu (mM)
**Pre-ischemia**					
Control	7.38 ± 0.03	83.6 ± 8.85	37.0 ± 3.92	105.2 ± 8.81	6.39 ± 1.23
Vehicle	7.39 ± 0.03	83.9 ± 8.53	36.7 ± 3.56	102.3 ± 8.35	6.34 ± 1.09
Control-PEBP1 (10 mg/kg)	7.43 ± 0.04	84.1 ± 9.10	37.2 ± 4.09	103.1 ± 9.02	6.42 ± 0.88
PEP-1-PEBP1 (1 mg/kg)	7.41 ± 0.04	83.4 ± 9.11	37.1 ± 4.55	104.2 ± 9.44	6.37 ± 1.03
PEP-1-PEBP1 (3 mg/kg)	7.43 ± 0.03	83.8 ± 8.95	36.8 ± 3.94	106.3 ± 9.21	6.48 ± 1.31
PEP-1-PEBP1 (10 mg/kg)	7.42 ± 0.04	83.5 ± 8.66	37.1 ± 4.45	106.9 ± 10.2	6.40 ± 1.07
**Reperfusion 10 min**					
Control	7.40 ± 0.04	84.8 ± 8.99	36.6 ± 3.96	107.4 ± 9.81	6.46 ± 0.98
Vehicle	7.37 ± 0.07	87.2 ± 10.2	39.4 ± 5.18	108.1 ± 9.60	6.93 ± 1.44
Control-PEBP1 (10 mg/kg)	7.35 ± 0.09	87.5 ± 11.3	38.8 ± 4.82	110.9 ± 10.8	7.05 ± 1.28
PEP-1-PEBP1 (1 mg/kg)	7.36 ± 0.06	88.2 ± 9.58	39.6 ± 6.02	106.5 ± 10.6	7.12 ± 1.36
PEP-1-PEBP1 (3 mg/kg)	7.37 ± 0.08	86.9 ± 9.83	38.4 ± 4.88	113.1 ± 12.2	7.07 ± 1.20
PEP-1-PEBP1 (10 mg/kg)	7.37 ± 0.10	85.2 ± 8.86	38.3 ± 5.03	109.8 ± 11.2	7.19 ± 1.24

There are no significant changes in physiological parameters in control, vehicle-treated, control-PEBP1-treated, and PEP-1-PEBP1-treated groups before ischemic surgery and 10 min after reperfusion.

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
