# Peer review of "Phosphatidylethanolamine-Binding Protein 1 Ameliorates Ischemia-Induced Inflammation and Neuronal Damage in the Rabbit Spinal Cord"

_cells, 2019, doi:10.3390/cells8111370_

Round 1
Reviewer 1 Report
The manuscript ” Phosphatidylethanolamine-binding protein 1 ameliorates ischemia-induced inflammation and neuronal damage in the rabbit spinal cord” by Woosuk Kim et al. looks specifically at the levels and role of PEBP1 against oxidative stress in NSC34 cells in vitro and ischemic damage in the rabbit spinal cord in vivo.
Authors found that treatment with PEP-1-PEBP1 fusion protein decreased the induction of oxidative stress in NSC34 cells and the following cellular death. Also, it did not show any significant side effects before and after ischemia-reperfusion. In addition, PEP-1-PEBP1 administration decreased oxidative stress and inflammation in rabbit spinal cord, and concluded that a possible PEP-1-PEBP1 treatment may have therapeutic value for patients with spinal cord ischemia.
I thought this was a well written and organized manuscript with a well-designed analysis, and the conclusions supporting the data. I congratulate the authors on the study. However, there are some issues that needs addressed.
First I would suggest that the authors go through the manuscript and shorten the long sentences comprising 3-5 lines, since reader can lose its breath and focus.
Abstract:
The abbreviations PEP and NeuN are used in the abstract; however, these are not explained. Please provide their full name first time they are mentioned in the manuscript.
Introduction:
The sentence in line 53-55 makes little sense in the context. Please provide reference to the statement “Neurons have many unsaturated fatty acids and utilize glucose as an energy source, and an interruption of blood supply and reoxygenation enormously increases the formation of ROS.” In line 61-62
Materials and methods:
Something is missing in sentence 87. Which conditions are you referring to? I wonder if data follows a normal distribution? Authors do not mention using any tests for checking the distribution, but assume it does and use parametric tests to describe the experimental data. When the sample size is small and especially when working with proteins, data does not necessarily follow a normal distribution, and hence, some of the results may be wrongly assumed. Please elaborate Section 2.4 and provide information about normality and possible log transformation. See more elaborated comments in the Results section regarding this issue.
Results:
I congratulate the authors for their work flow through this section. Is it possible to change the yellow colour in Figure 4A (PEP-1-peptide) since the line is not visible? In all figures and results authors use means and standard error of means (SEM). While using SEM provides nice plots, they do not necessarily show the real picture, especially when the number of samples is only 5 or 10 per group. A more correct way to show the results is through means and standard deviations/95%CL, box-plots, and/or scatter plots where every sample is shown. Please consider the presentation of these figures based on data distribution.
Discussion:
This section needs to be more differentiated from the Results section, since several sentences describing the findings are nearly re-written from the previous section.
Once again, I congratulate you for the nice study and well-written manuscript and I am looking forward to reading the new version.
Author Response
First of all, I apologized the late submission of revised manuscript and I appreciated the reviewer’s prudent comments for our manuscript. The responses to the reviewers’ comments are summarized below and highlighted with red color in the manuscript:
Reviewer #1
Comment 1. First I would suggest that the authors go through the manuscript and shorten the long sentences comprising 3-5 lines, since reader can lose its breath and focus.
Answer 1. We thoroughly edited the manuscript with the help of our colleagues and one native speaker to shorten the long sentences according to reviewer’s comment. Please refer to the text.
Abstract:
Comment 2. The abbreviations PEP and NeuN are used in the abstract; however, these are not explained. Please provide their full name first time they are mentioned in the manuscript.
Answer 2. We showed the full name (neuronal nuclei) of NeuN in the abstract and method sections according to reviewer’s comment. PEP-1 peptide is not an abbreviated word. Please comprehend this.
Introduction:
Comment 3. The sentence in line 53-55 makes little sense in the context. Please provide reference to the statement “Neurons have many unsaturated fatty acids and utilize glucose as an energy source, and an interruption of blood supply and reoxygenation enormously increases the formation of ROS.” In line 61-62
Answer 3. We cited a reference to support the statement according to reviewer’s comment and refer to the text.
Materials and methods:
Comment 4. Something is missing in sentence 87. Which conditions are you referring to? I wonder if data follows a normal distribution? Authors do not mention using any tests for checking the distribution, but assume it does and use parametric tests to describe the experimental data. When the sample size is small and especially when working with proteins, data does not necessarily follow a normal distribution, and hence, some of the results may be wrongly assumed. Please elaborate Section 2.4 and provide information about normality and possible log transformation. See more elaborated comments in the Results section regarding this issue.
Answer 4. We make a mistake and we cultured the NSC34 cells according to methods described in the previous study. We shortened the sentence because of self-plagiarism. Please comprehend this point. For statistical analysis, we fully agreed reviewer’s comment. We assumed that the experimental data follow normal distribution in control and vehicle-treated groups because we and our colleagues repeatedly conducted the experiments (more than 300 in each group) using in vitro oxidative stress model in NSC34 cells and in vivo spinal cord ischemic model in rabbit. PEP-1-PEBP1-treated group did not follow the normal distribution, but we can predict the neuroprotective potentials of experimental groups by comparing the normal distribution patterns of control and ischemia group.
Results:
Comment 5. I congratulate the authors for their work flow through this section. Is it possible to change the yellow colour in Figure 4A (PEP-1-peptide) since the line is not visible? In all figures and results authors use means and standard error of means (SEM). While using SEM provides nice plots, they do not necessarily show the real picture, especially when the number of samples is only 5 or 10 per group. A more correct way to show the results is through means and standard deviations/95%CL, box-plots, and/or scatter plots where every sample is shown. Please consider the presentation of these figures based on data distribution.
Answer 5. We edited the yellow color in Figure 4A into dark grass color. Because of very similar results with Control-PBEP1 (10) group, it is only noted at 72 h after ischemia. We modified the Figure 5 with standard deviations/95%CL scatter plots according to reviewer’s comment. Please refer to the Figures 4A and 5.
Discussion:
Comment 6. This section needs to be more differentiated from the Results section, since several sentences describing the findings are nearly re-written from the previous section.
Answer 6. We tried to differentiate the sentence from result section and some sentences were edited. Please refer to the text.
Reviewer 2 Report
Spinal cord ischemia causes oxidative stress and subsequent inflammation, leading to neuronal injury and motor neuron loss. In this article, the author discovered that delivery of phosphatidylethanolamine-binding protein 1 (PEBP1) reduces neuronal death caused by oxidative stress and spinal cord ischemia. In addition, they show that PEBP1 reduces the levels of oxidative stress biomarkers MDA, AOPP and 8-iso-PGF2a, as well as the level of proinflammatory mediators MPO, TNF-a and HMGB, suggesting that PEBP1 may serve as a therapeutic approach for spinal cord ischemia. However, there are some questions to be addressed before accepted for publication.
In Figure3C, the author used TUNEL assay to evaluate the role of PEBP1 in rescuing oxidative stress induced cell death. Additional measurement such as caspase3 should be included to assess apoptotic cell death. In addition, the quantification of TUNEL assay should be calculated by TUNEL positive cells per mm2normalized to DAPI staining number, but not just measure fluorescence intensity.
In Figure4, the authors stated that PEBP1 delivery increase neuron viability using NeuN staining. However, single NeuN staining cannot determine the status. TUNEL or Caspase3 staining should be included together with NeuN staining to determine whether PEBP1 can rescue neuronal damage after spinal cord ischemia.
The author need to show the evidence that PEBP1 is efficiently delivered in vivo.
The Figure3A is not labeled clearly and it is very difficult to understand. What is the unit for 1, 2, 3? What is the difference between vehicle and H2O2treatment?
In Figure5, the authors showed that PEP1-PEBP1 decreases ischemia-induced upregulation of MDA, AOPP, 8-iso-PGF2a, MPO, TNF-a and HMGB1. However, the authors need to include control-PEBP1 as an additional control group.
Author Response
First of all, I apologized the late submission of revised manuscript and I appreciated the reviewer’s prudent comments for our manuscript. The responses to the reviewers’ comments are summarized below and highlighted with red color in the manuscript:
Reviewer 2.
Comment 1. In Figure3C, the author used TUNEL assay to evaluate the role of PEBP1 in rescuing oxidative stress induced cell death. Additional measurement such as caspase3 should be included to assess apoptotic cell death. In addition, the quantification of TUNEL assay should be calculated by TUNEL positive cells per mm2 normalized to DAPI staining number, but not just measure fluorescence intensity.
Answer 1. We fully agreed the reviewer’s comments and we newly conducted western blot analysis for naïve and cleaved caspase-3 levels in the cells and we showed the quantitative data in the Figures 3. Please refer to the text and Figure 3.
Comment 2. In Figure4, the authors stated that PEBP1 delivery increase neuron viability using NeuN staining. However, single NeuN staining cannot determine the status. TUNEL or Caspase3 staining should be included together with NeuN staining to determine whether PEBP1 can rescue neuronal damage after spinal cord ischemia.
Answer 2. We conducted the Fluoro-Jade C staining to detect the damaged cells in the spinal cord and we showed this data in Figure 4. In addition, we also measured the naïve and cleaved caspase-3 levels by western blot analysis. Please refer to the text and Figure 4.
Comment 3. The author need to show the evidence that PEBP1 is efficiently delivered in vivo.
Answer 3. In the previous studies, we observed the efficient delivery of PEP-1-cargo (PEP-1-heme oxygenase-1) proteins in the brain by immunohistochemical staining for polyhistidine antibody (Jung et al., Neurochem Res. 2016 Apr;41(4):869-79). We cited the reference according to reviewer’s comment. Please refer to the text.
Comment 4. The Figure3A is not labeled clearly and it is very difficult to understand. What is the unit for 1, 2, 3? What is the difference between vehicle and H2O2 treatment?
Answer 4. I appreciated your comments and it means the concentration of H2O2 added to induce the oxidative stress. To give clear information, we inserted “concentration” in the Figure 3A. Please refer to the Figure 3A.
Comment 5. In Figure 5, the authors showed that PEP1-PEBP1 decreases ischemia-induced upregulation of MDA, AOPP, 8-iso-PGF2a, MPO, TNF-a and HMGB1. However, the authors need to include control-PEBP1 as an additional control group.
Answer 5. We added the control-PEBP1 (10) group for measurements of MDA, AOPP, 8-iso-PGF2a, MPO, TNF-a and HMGB1. We described the results according to reviewer’s comment. Please refer to the text and Figure 5.
Round 2
Reviewer 1 Report
Thank you for meeting the requested issues. I have no further comments.
Reviewer 2 Report
None